# Mainstreaming Gender into Irrigation: Experiences from Pakistan

**Junaid Alam Memon** [1,*] **, Bethany Cooper** [2] **and Sarah Wheeler** [3]

[1] School of Public Policy, Pakistan Institute of Development Economics, Islamabad 44000, Pakistan
[2] School of Commerce, University of South Australia, Adelaide, SA 5001, Australia; bethany.cooper@unisa.edu.au
[3] Centre for Global Food and Resources, The University of Adelaide, Adelaide, SA 5005, Australia; sarah.wheeler@adelaide.edu.au
[*] Correspondence: junaid.alam@alumni.ait.asia; Tel.: +92-333-5271153

**Abstract:** The influence of gender in participatory irrigation management reforms has been the subject of significant research in the past. Whilst there is some understanding of what hinders women and marginalized groups from participating in irrigation management, there is limited understanding of how male and female farmers vary in their perceptions on the effectiveness of participation in irrigation affairs. There is also limited understanding around the interaction on gender and the overall success of participatory irrigation management programs. Based on the information obtained from 128 households surveyed through separate male and female questionnaires in Pakistan in 2018 (Sindh and Punjab provinces), we studied the country's experience in engaging gender into its participatory irrigation management program. We found there was a significant difference in participatory irrigation management perceptions across both gender and locational jurisdiction. Overall, women generally perceive the performance and impact of farmer organizations to be significantly less effective than men. Our study emphasizes the importance of putting findings in a historical context to inform the theory, policy, and practice of mainstreaming gender into irrigation management.

**Keywords:** gender analysis; inclusion; participatory irrigation management; farmer organizations; water user associations; Indus Basin irrigation system; irrigation and drainage authority

## 1. Introduction and Background

For many developing countries of Asia and Africa, where a majority of the population is engaged in agricultural production, the participation of women in agricultural activities is an important area of development studies. In addition, irrigation has often been described as an important focus needed for further agricultural development in developing countries, though studies indicate that in general there is more net economic benefit from small-scale rather than large-scale irrigation development [1]. Increasingly, from the 1980s onwards, there has been research conducted on the merits of participatory irrigation management [2,3], and the literature has also sought to highlight the issues of women's participation in irrigation management in particular [4–7].

These influential studies highlighted not only the lack of women's access to irrigation but critically evaluated the disciplinary biases, socio-cultural issues, and political economy and described the irrigation sector as a highly technical male domain that systematically nullifies women's participation. In doing so, studies highlighted challenges, such as: (i) Men's and women's differential bargaining powers and preferences [8]; (ii) irrigation policy and practice that overlooks the role of women [9]; and (iii) the intra-household and intra-community power gap and its implication for irrigation equity and effectiveness [10].

These studies identify women's participation and empowerment as very important, cutting across multiple themes of social, economic, and sustainable development. Participation is also seen as a development outcome. With reference to water, for example, the Dublin Statement's Principle No. 3 acknowledged women playing a central part in the provision, management, and safeguarding of water [11]. Kevany and Huisingh [8] suggest that removing barriers to women's participation and empowerment unleashes their latent potential to raise human progress and wellbeing in general and improve irrigation outcomes in particular.

A plethora of empirical evidence also suggests that equal participation of women leads to healthier and richer societies and economies [12]. These understandings and subsequent international and national policy efforts have brought significant improvements in female participation. Yet, some 100 countries have a way to go in ensuring women's equal access to legal and economic rights, such as an equal share in inheritance, capability to sign a contract and manage a property, interaction in public spheres, meaningful engagement in labor markets, travel abroad, and other important domains [13,14].

The conceptual link between gender and participation in water management can be drawn in many distinct ways. To start with, the concept of 'gender' has been discussed in varying ways over time and has been subject to much debate [15–17]. This conceptualization includes 'women in development' during the 1980s, to 'gender and development' in the 1990s [18], and then progressively as 'gender mainstreaming'. Such a progression illustrates that 'gender' as a concept and focus in the literature has transformed substantially, but we have not seen a corresponding transformation in the lives of women in developing countries [19]. Anouka and Tine [20] argue that disappointment with 'gender mainstreaming' has more to do with the process of its institutionalization than the concept itself.

Similarly, 'participation' also suffers from the ease of institutionalization, depoliticization, and sanitization from its original transformative potential [21]. Conceptually speaking, although participation of marginalized groups (such as women) is appropriation of power by the powerless, participation in reality may mean anything from genuine involvement to manipulation and citizen control—the so-called ladder of citizen participation [22]. Notably, there is limited empirical evidence to suggest that citizens' participation exhibits a natural progression from the lowest to the highest rung of the ladder.

Against this backdrop, this study is interested in exploring this issue further for irrigation regions in Pakistan. Previous research on South Asian women's participation in water management has often been drawn based on their visibility in the spaces observable to researchers and development practitioners rather than a detailed understanding of women's actual influence on water management decisions [17,23]. Although the requirement and policy prescriptions on gender involvement and water may tend to increase women's visibility in water management spaces, the depth of their real involvement remains unknown.

Some examples of the incentives that have been put in place to promote the participation of women in irrigation include designing infrastructure that caters to women's water needs (e.g., washing ghats along irrigation channels), forming women water groups, and including a woman on a water management committee [9,24]. However, the issue with such incentives is that women often do not have any legal right to influence water allocation decisions. Thus, even if women are invited onto water management committees, their participation may be limited due to structural inequalities and a lack of exposure and confidence [10]. In addition, an unconventional entry into the "male domains" may trigger serious conflicts and endanger the participatory irrigation management agenda itself [10]. However, on the other hand, it can also create possibilities to renegotiate gender norms, roles, and relations in patriarchal societies of South Asia in general and Pakistan in particular [17].

Although the recent literature has provided rich insights based on in-depth qualitative analysis of gender and irrigation in different parts of South Asia, including Pakistan [25], there is limited quantitative analysis of gender-based differences in the perceptions regarding participatory irrigation management (PIM) systems and their performance. This paper attempts to bridge this gap based on data assembled from PIM experiences in the Sindh and Punjab provinces in Pakistan.

In the current world, the involvement of women and their values and opinions is seen as important [11]. This covers multiple domains and is progressively creeping into masculine areas of influence, like agriculture [7,8]. However, despite the perceived nobility of inclusion, its perceived benefits on the part of those 'not being included' is not always clear cut [7]. More specifically, there is a presumption that those excluded hold several views, such as: (a) That 'things' would be better if the excluded had a voice, (b) the excluded see the same benefits of the 'included' and want to share the spoils, and (c) the included and excluded hold similar values and perceptions. This paper explores the extent to which these assumptions hold in the policy and legislative landscape in Pakistan where the world's largest gravity-driven irrigation system operates from the Indus River.

## 2. Policy and Institutional Context of Women's Economic and Political Rights in Pakistan

### 2.1. Historical Account of Gender Mainstreaming

Female participation in Pakistani society has been shaped by various historical and political factors. Historically many struggles against suppression and discrimination have been focused on land rights and taxation. The Mughal Empire in the Indian subcontinent (1526 to 1540 and 1556 to 1850s) employed oppressive elite systems of Mansabdari for their civil and military administration. In their latter period, these systems triggered peasant revolts throughout India (triggering leadership of local heroes and heroines, including women, such as Mai Laddhi (Punjab), Mai Bakhtawar (Sindh), and Bibi Alai (Pakhtun areas) [26].

However, a more recent account of women's economic and political rights struggle can be traced in the British era from the 1850s onwards. Though the British adopted a rhetoric of modernity in many areas, they did not uproot any traditional institutions (including patriarchy) whenever it suited their interests [27]. Initially, the British abolished women's right to inherit property that was previously allowed in Islam; but later on, the promulgation of Muslim Personal Law 1937 bestowed females the right to inherit parental property other than agricultural land [27]. Interestingly Muslim women's political engagement was strategically exploited by indigenous leaders for various national, social, and religious causes but was discouraged in struggles (e.g., the educational reforms movement) that may have enabled women to self-reflect and renegotiate traditional gender roles. Despite reluctance on behalf of the British rulers and indigenous leaders, Muslim women obtained the right to vote, attend modern education, and inherit property (other than agricultural land), primarily because their numbers had strategic importance in Muslim minority politics in India [27,28]. However, the major drawback of women struggles during this era was mostly urban-based, elite centric, and indifferent to rural women issues [29].

British rule ended in 1947, splitting the subcontinent into India and Pakistan. The pre-partition opportunity to effectively engage in the nationalist and religious movements raised women's political awareness and action [30]. Starting at Pakistan's first national budget discussion in 1948, women lodged a long struggle for political and economic rights recognition, comprising of strikes, protests, and demonstrations. They won the right to inherit agricultural land (bestowed in the Muslim Personal Law of Shariat (1948) [28], dual suffrage for women to vote for general, and reserved female seats in national and provincial assemblies (agreed to in the 1956 Constitution). However, with the abolition of the 1956 Constitution due to a military coup in 1958, the legislation of women's suffrage against their reserved female was repealed and never restored [27,28,30].

Since religion legitimized the creation of Pakistan, it enables religious zealots to exercise great influence over state affairs. Even a socialist like Bhutto frequently clarified that his political ideology was not un-Islamic and to demonstrate this, he strengthened the religious dominance in the state affairs by framing the Constitution of 1973 on the principle of 'Islamic Socialism'. Unfortunately, the religious clerics in Pakistan who had been the custodian of oppressive patriarchal customs held this up as Islamic. Clerics never approved Bhutto partially because of the nominal improvement he made in female rights [31]. Bhutto was overthrown and ultimately hanged by General Zia's notorious martial

law, who authored what is certainly the darkest chapter in the history of women's economic and political rights legislation in Pakistan. This included: (i) The Hudood Ordinances (1979) that abridged the distinction between rape and adultery, which victimized women for being raped while men walked away with impunity and was frequently used as a terrorizing tool by husbands [32]; (ii) the Qisas and Diyat Ordinance that privatized women's murder in the name of honor; and (iii) the Law of Evidence that made women testimony inferior and half of that of men [32]. Attempts were also made to repeal the Family Laws Ordinance of 1961 (that protected women rights after marriage) and their right to serve on important public offices, travel abroad without a legal male companion (husband, brother, and father), and their right to participate in sports [27,32].

Women organizations, such as the All-Pakistan Women's Association, Women's Action Forum, Sindhiani Tehrik, and Women's Front and Democratic Women's Association, all condemned General Zia's discriminating policies but were successful in saving only a few victims [32]. Even four democratic regimes (1988–1999) following General Zia's era, two of which were led by a female prime minister (Benazir Bhutto), could not repeal the anti-women legislation. To some extent, General Pervaiz Musharraf (who installed himself as Pakistan's president through a military coup and reigned during 2001–2006) window-dressed women's concerns through the Women's Protection Bill 2006 [27,28]. During the eleven years of his reign (1978–1988), Zia and his fellow clerics cultivated intolerance to an extent that their legacy has survived, and women feel unsafe even today.

Amid this, the global development narratives also saw a major shift upon the end of the cold war that established the supremacy of the capitalist bloc. It effectively replaced revolutions with reform and installed private sector and non-governmental organizations (NGOs) as an alternative, and adjunct to, governments [18]. Like elsewhere in the world, many women's movements in Pakistan also buried their political ideologies to embrace apolitical ideas, which later matured as 'gender and development' and 'gender mainstreaming'. While narrating the story of Pakistan, Saigol [27] describes how the struggles of various women turned into a nine-to-five NGO job. Meanwhile, the 1973 Constitution has been amended 18 times and women reserved seats in the national and provincial assembly make 17.5% of the total in 2019. However, these are still filled through political parties' nominations in proportion to the number of general seats they win instead of women's suffrage. Women who are selected on these seats belong to privileged groups—as had been the case through most of the country's history (see [32]). Currently, women's cause is at a juncture where its future depends on how effectively they engage with the state that has historically denied them their fundamental human, economic, religious, and political rights but at the same time is still at risk of going backwards [27].

### 2.2. Provincial Account of Women's Rights Struggle

As Hamza Alavi [32] described, the set of problems that different classes of women face in patriarchal society needs different strategies for resolution. Although elite women associations generally have been very prominent, they were concentrated in urban areas, such as Lahore, Karachi, and Peshawar, and had an insignificant understanding of the life of rural women [26,29]. To resolve the difficulties of Sindhi rural women, which increased many fold in the Zia regime, the Sindhiani Tehrik (ST) emerged from the lower Sindh's Thatta and Badin districts and has the privilege of being the "first ever progressive and exclusive women organization in the history of sub-continent" [33]. The ST is the only noteworthy female movement that was grassroot, peasant-focused, and acted as a rural women's voice against patriarchal and oppressive institutions and actions. It relied on songs and poetry for mass mobilization for change and believed in direct action against all kinds of oppression and discrimination, including equal wages, bonded labor by landlords, and rape [34].

Within a couple years of its formation in 1981, ST mustered peasant and laborer women's support from large parts of Sindh [27]. ST's role in the Movement for Restoration of Democracy (MRD) against Zia's tyranny was so prominent that it was declared to be a genuine threat to General Zia. Though MRD ended with the death of General Zia, ST continues to struggle for peasant and laborer women's rights and has organized various successful mass mobilizations on the issue of provincial water rights,

military operations in Baluchistan, and other instances of violence against marginalized sections of society [33].

Unfortunately, a similar kind of exclusive women's organization did not emerge in Punjab, though there are instances when women came out of their homes and joined their male counterparts in the struggle against unjust actions of the state. Salim Ahmad [26] has documented many instances in history where Punjabi women have also stood up against oppression and discrimination.

## 2.3. Evolution of the Indus River's Irrigation Network

The subject location of this study, the Indus River, has experienced extraordinary civil works over the last one and a half centuries (Figure 1). Though the state sponsored irrigation works were carried out even before the British rule, those were not comparable to developments that took place afterwards. Starting with the Sirhind Canal in 1882 and the Sidhnai weir in 1886, the British started a new era of irrigation engineering domination over the mighty Indus River. Other major infrastructure commissioned by the British included: Khanki Headworks and Rasul Headworks till 1900; Upper Chenab Link Canal, Balloki Barrage, Upper Jhelum Link Canal, Mangla Headworks, Marala Barrage, Islam Barrage, and Sulemanki Barrage during the 1910s and 1920s; and Panjnad Barrage, Sukkur Barrage, Trimmu Barrage, and Jinnah Barrage after the 1930s, until the end of British rule and partition of the subcontinent in 1947. Between 1947 and 1960, despite ambitious plans, irrigation development remained somewhat restricted due to the post-partition conflict between the newly created India and Pakistan. However, the major works commissioned in Pakistan during this period were: Balloki Sulemanki Link canal, Kotri Barrage, Marla-Ravi Link Canal, Bambawli-Ravi-Bedian Link Canal, Taunsa Barrage, and the first phase of the Warsak Dam [35].

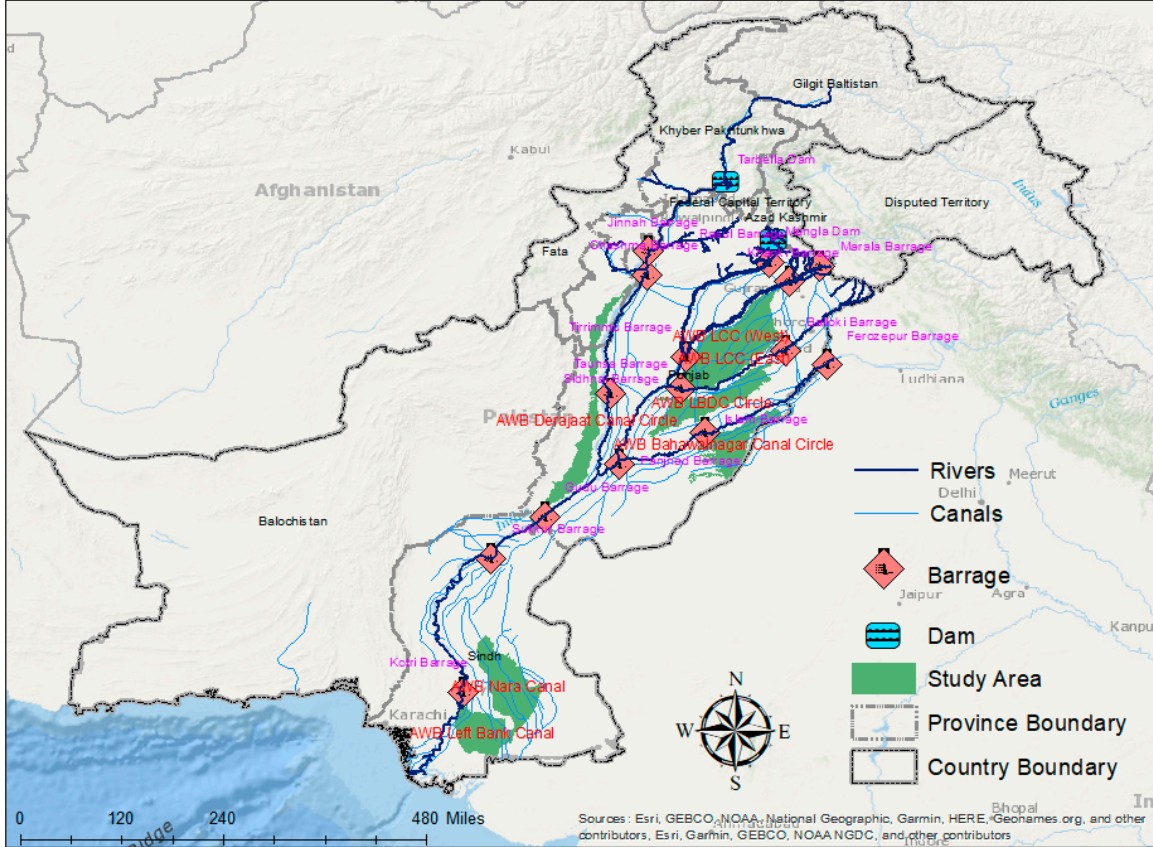

**Figure 1.** Irrigation development on the Indus River and location of the study area. Source: Thematic layers by authors; base map from the sources mentioned inside the map.

In 1960, India and Pakistan signed the Indus Waters Treaty that was essentially a dissection of the Indus River system. Subsequently, exclusive rights of India were held over the Sutlej, the Beas, and the Ravi rivers (~20% of the Indus Waters) and relatively exclusive rights of Pakistan were established over the Indus, the Jhelum, and the Chenab rivers (~80% of the Indus waters). The decade that followed facilitated various major irrigation works which included: Guddu Barrage, Sindhiani Barrage, Sidhnai–Maisli Link Canal, Malisi–Bhawal Link Canal, Trimmu–Sidhnai Link Canal, Siphon Barrage, Rasul–Qadirabad Link Canal, Mangla Dam, Qadirabad Barrage, Qadirabad–Balloki Link Canal, and Rasul Barrage. From 1970 to the 1990s, the system was augmented with Taunsa–Panjnad Link Canal, Chasma–Jhelum Link Canal, Chasma Barrage, and Tarbella Dam and its extension. Since then, although water resources development has slowed down due to a lack of investment and inter-provincial conflict, the infrastructure commissioned includes Thal Canal, Katchi Canal, and Kurramtangi Dam [35].

Today, the Pakistani side of the Indus River houses the world's largest contagious irrigation system built with an estimated investment of about USD 300 billion [35]. It consists of three gigantic dams, 23 barrages, 12 inter-river link canals, 48 perennial and non-perennial canals, and hundreds of thousands of watercourses, irrigating about 14 million hectares of agricultural land [36]. The system is the mainstay of Pakistan agriculture, with agriculture in 2018 contributing around 18.5% to the country's gross domestic product (GDP) and employing 38.5% of the national labor force [37].

## 2.4. Water Resources Management in the Indus River Basin

Although pre-partition irrigation investments in the subcontinent served colonial motives, those done post-partition in Pakistan were justified as a catalyst of socioeconomic change [38]. By the 1970s, it was very clear to the government and the World Bank that the system was only able to recover a small fraction of its operation and maintenance (O & M) costs. The large landlords who own the majority of the agricultural land in Pakistan sought to influence all levels of irrigation affairs. For example, this included policies subsidizing water charges in the name of smallholders, to the position of watercourse regulator gates to appropriate a larger share of the available water [39–41]. By the early 1980s, the Indus irrigation system was failing badly due to poor O & M arising out of unjustified subsidies, low crop assessment and cost recovery, inequitable irrigation distribution, and widespread corruption and inefficiencies [40,42–44].

During this time, PIM was perceived as a panacea to similar problems in many countries. The World Bank pushed Pakistan, like many other countries, to transform its top-down irrigation management into a more participatory and farmer-managed system. Years of action research, dispute, and policy dialogue resulted in the Provincial Irrigation and Drainage Authorities Act in 1997. Subsequently autonomous institutions of irrigation management were established at different levels: Sindh Irrigation and Drainage Authority in Sindh and Punjab Irrigation and Drainage Authority in Punjab were mandated to manage provincial-level irrigation affairs; whereas, under each of these entities, there were Area Water Boards (AWBs), which managed canal circle-level issues. Each AWB had various tertiary-level Farmer Organizations (FOs), where farmers were organized into Water User Associations (WUA) and Drainage Beneficiary Groups (DBGs) [43,44].

Meanwhile, the Sindh Irrigation and Drainage Authority Act 1998 was replaced with the Sindh Water Management Ordinance (SWMO) 2002 [45] and further amended as the Sindh Water Management (amendment) Act 2005 [46]. These legislations gradually minimized the bureaucratic influence of engineers, as well as declaring FO, AWB, and SIDA as separate entities whose interrelations were to be governed by a Regulatory Authority—a role temporarily bestowed to SIDA. Since 2017, NGOs like Strengthening Participatory Organizations (SPO) and Oxfam have advocated for further amendments to the SWMO to increase female and other marginalized groups' inclusiveness [47]. On the contrary, the Punjab Irrigation and Drainage Authority Act 1998 [48] persisted in its original form until it was repealed with the Punjab Khal Panchayat Act 2019 [49]. This legislation reverted the already limited rise of participatory irrigation institutions in Punjab. Seemingly, it has retained only watercourse-level associations but has made these subservient to irrigation bureaucracy [49].

*2.5. Legislative and Policy Support for Women's Participation in Water Resources Management*

Though there is a considerable documented account of women's struggle for land rights in Pakistan, the same is not true for women's participation in water resources management and irrigation. Despite irrigation and its related affairs being subject to important changes in legislation in the past, women were not perceived as a stakeholder in this domain until the late 1990s (Figure 2). At least two plausible explanations can be offered for this: first, water rights in Pakistan are attached to land [50], and the struggle of women for land rights imply that water rights will follow. Thus, there seems no need for them to have lodged a separate struggle for water rights. Second, since the Mughal era, irrigation development on the Indus River system required large-scale public works and it was almost always beyond the capacity of local communities to initiate any major irrigation works. This was reinforced during the British era, where, as described previously, state-of-the-art engineering works were initiated [36]. The British achieved this by developing a cadre of hydraulic bureaucracies that, like elsewhere, tend to be dominated by masculine custodians of technologies, such that irrigation technology in Pakistan is still perceived as a 'man's world' [39,51] and virtually no woman occupies any technical position. Thus, it is perhaps understandable why women have had a lack of participation in water management over time.

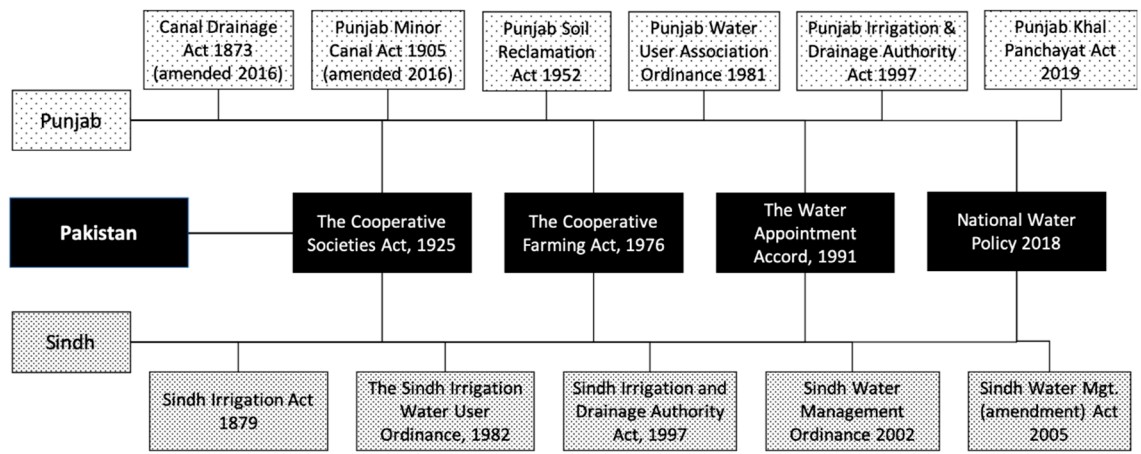

**Figure 2.** Irrigation policy and legislation in Pakistan. Source: Authors.

Since participatory irrigation reforms over the last three decades have been a donor-driven move, gender participation was made compulsory. However, the initial legislation (i.e., Sindh and Punjab Irrigation and Drainage Authority Acts) lacked any consideration for females since only those farmers who owned or rented the land could join farmer organizations under participatory irrigation systems. Since very few females own land, there was no scope for their participation. However, to show 'inclusiveness', the irrigation and drainage authorities formed women groups without any formal say in farmer organizations' management and decision-making [9,10]. Empirical studies on participatory irrigation management in Sindh show that rarely any women serve on the management committees of farmer organizations [9,52]. Although the Sindh Water Management Ordinance 2002 had a few clauses on women participation in farmer organizations, no serious consideration was given to women's active role in managing irrigation. As a result, even today (e.g., see National Water Policy 2018 [53]), any consideration of women tends to be concentrated upon water for domestic needs only (Figure 2).

In sum, this historical account suggests that the legislative and policy support for women's participation in water management is extensive and multifaceted. The complexity is heightened by the socio-cultural and political environment. Whilst there have been numerous half-hearted efforts to improve the involvement of women in water management, it is evident that there is still a lack of understanding of female preferences.

This study seeks to understand these issues in more detail, by exploring male and female perceptions of participatory management in two irrigation regions in Pakistan. It also provides insights into the role of gender in the context of decision-making in irrigation.

### 2.6. Allocating Resources: Household Decision-Making

There is a significant body of literature that recognizes intra-household inequality and contributes to an understanding of how households allocate resources [54–56]. There are a number of established indices to capture and measure inequality within the household [57]. There is often a focus on measuring control in two ways: first, as control over the management of household finances and, second, as an influence over household decision-making [57]. Notwithstanding that intra-household inequality is pervasive across both agrarian and industrialized countries, here, we focus on this type of measurement in the context of developing nations [58–60]. Important policy concerns are intrinsically related to intra-household inequality. Any evaluation of women's, men's, and children's relative access to services, goods, and leisure must be underpinned by some assumption about how resources are allocated within households. Moreover, it is evident that women have a greater propensity to spend on children than men. For example, mothers' access to income is a more crucial determinant of children' health and nutrition than the total household income [57,61].

An apparent trend in the literature highlights the importance of women having comparable access to resources to men. More specifically, the ability of women to access resources, such as household finance, education, land, credit, and water, reduces inequality [62]. In a similar vein, it is suggested that greater inclusion of women in water management can also assist in addressing gender inequality. Moreover, some argue that the management of water itself stands to be improved by greater inclusion. In essence, participatory irrigation seeks to improve the involvement and cooperation of key stakeholders. Given water is a common pool resource, finding cooperative solutions for use and sharing are more efficient ceteris paribus. Arguably, women may engender more cooperative outcomes than men, although this is of course subject to context [63,64], but this can make it easier (and less costly) to manage water conflicts.

Reducing gender inequalities and empowering women are two central objectives of the global development debate and policy. Women's empowerment and gender inequality are usually measured at the aggregate country level; thus, it overlooks heterogeneity between regions. The Women's Empowerment in Agriculture Index (WEAI) was developed to measure the empowerment, agency, and inclusion of women in the agricultural sector [65]. Although there are five domains covered by the WEAI, in this study, we focus solely on decisions about agricultural production. The following provides more details of our method.

## 3. Materials and Methods

This study is part of a larger project that is funded by the Australian Centre for International Agricultural Research (ACIAR): Efficient Participatory Irrigation Institutions to Support Productive and Sustainable Agriculture in South Asia. Among others, the project undertook two surveys in 2018 in irrigation regions in Sindh and Punjab. The irrigation regions included from Sindh were: Nara Canal Area Water Board (AWB) and Left Canal AWB; and the irrigation regions included from Punjab were: Lower Chenab Canal (East) Circle AWB, Lower Chenab Canal (West) Circle AWB, Lower Canal Circle Bahwalnagar AWB, Lower Bar Doab Canal AWB, and Derajab Canal Circle AWB (see Figure 1). The Farmer Institutional Survey was completed by male respondents (survey One) and a Participatory Irrigation Survey was completed by female respondents (survey Two). Both surveys included several overlapping questions to capture respondents' perceptions on the performance and impact of farmer organizations. We extracted data for individual households who were covered in both surveys (hence representing both male and female heads of a household), with 128 household observations available. Forty-nine pairs were from Sindh (38% of the total sample), and 79 pairs were from Punjab (62% of the sample). Our motivation was to understand if there were significant differences in perceptions across

men and women farmers. In particular, we sought to develop a real understanding of the preferences of women as opposed to what men think women prefer.

A range of socio-economic household data, irrigation farm detail, irrigation institutions, perceptions, and attitudinal data were collected in the surveys. As previously mentioned, in particular, survey two employed a reduced form of the WEAI [65] to gain empirical insights into women's perceptions of their involvement in farm household decision-making related to irrigation. The response format required respondents to indicate who makes particular decisions using one of three mutually exclusive categories, namely: Men, women, or a joint decision. The irrigation decisions were framed as statements, such as: What source of irrigation water is used (canal/wells/others); when are crops irrigated; and spending of additional income generated from agriculture.

As stated above, survey one was administered to male farmers, whereas survey two was administered to female members of the farm households. In Pakistan, women exhibit not only restricted mobility in the public sphere but are also closely monitored for their interaction with outsiders, including female enumerators. Accordingly, survey two captured data regarding the presence of males at the time of data collection. On average, more than 87% of the in-person surveys were conducted only with female respondents, around 8% in the presence of one or more females and the remaining few in the presence of male family members.

Looking provincially reveals that in Punjab, all women interviews took place in strictly private settings; whereas in Sindh, about 65% of the interviews were conducted in strictly private settings, 20% in the presence of other women family members, and 14% in the presence of both male and female family members (Figure 3 provides an overview).

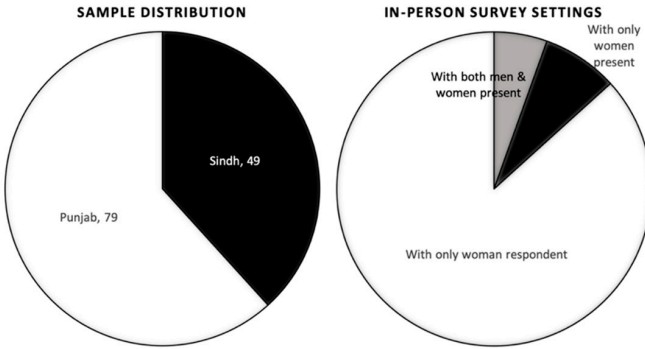

**Figure 3.** Sample distribution and in-person survey settings of gender survey (survey two).

Among other sample characteristics, the reported age range of female respondents was between 22 and 65 years, with an average of 41 years. Men were aged between 20 and 85 years, with an average age of 45 years (Table 1). Reported years of schooling for females were between 0 and 16 years, with an average of 4.1 years; whereas men schooling was between 0 and 17 years, with an average of 7.9 years. Total family land holding ranged between 0.5 and 100 acres—averaging at 12.3 acres; whereas females land holding ranged between 0 and 40 acres—averaging at 0.6 acres. However, more than 90% of females were landless (88.6% in Punjab and 93.9% in Sindh). Of the remaining females who owned land, the average landholding size was just 3.1 acres in Punjab and 14.6 acres in Sindh, owing to generally larger landholdings in Sindh compared to Punjab.

**Table 1.** Respondents' profile from the paired households (n = 128).

| Characteristic | Female (Survey 2) | Male (Survey 1) |
|---|---|---|
| Age of the respondent (years) | 40.9 | 44.6 |
| Years of schooling | 4.2 | 7.9 |
| Respondent as household heads (%) | 0 | 86.7 |
| Respondents as spouse to household head (%) | 88.3 | 0 |
| Land holdings (acres) | 0.6 | 12.3 |

## 4. Results

### 4.1. Involvement of Women in Household Decision-Making

As indicated previously, access to resources and decision-making power plays an important role in influencing the extent of gender inequality. Ten survey questions to capture the role of men and women in decision-making were adapted from the WEAI [65] and included in survey two. Figure 4 illustrates the role of men and women in irrigation decisions. The results reveal that none of these types of decisions are made solely by women. Alternatively, the majority of these decisions are made solely by men and a small portion are made jointly by men and women. Notably, no decisions are made jointly in Punjab.

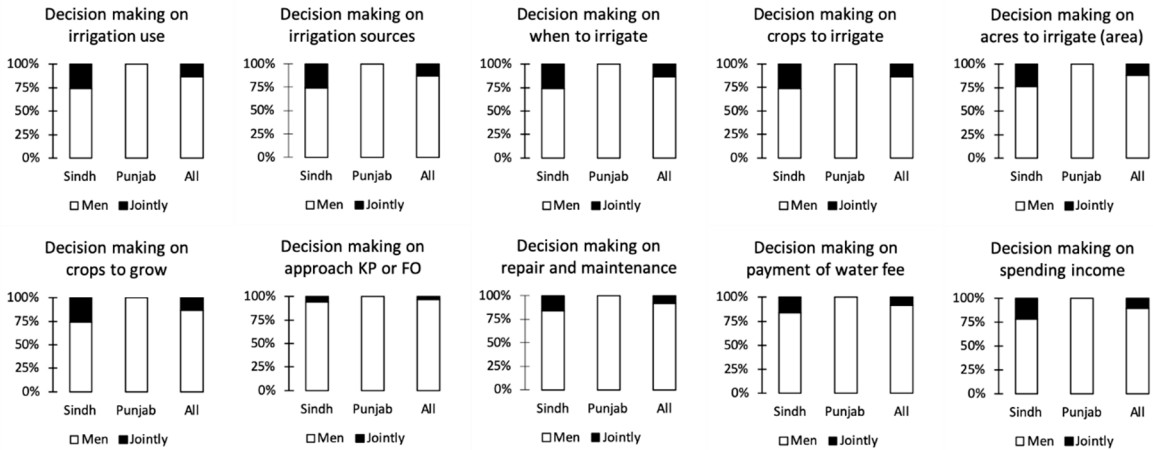

**Figure 4.** Decision-making in irrigation.

### 4.2. Rationalities for Participation in Irrigation Management

We sought to understand in detail the perceptions of men and women towards various reasons and rationalities for participation in irrigation management. Eight rationalities and questions were created and asked of respondents in both surveys (Table 2). A five-point Likert scale response format was used, where 1 = strongly disagree, 2 = disagree, 3 = undecided/neutral, 4 = agree, and 5 = strongly agree. A paired sample t-test was applied to infer statistically significant mean differences across gender perceptions on various aspects of PIM institutions. Each rationality index is the average of the sum of two questions.

Generally, households (which is the average of the male and female responses) tend to be either indecisive or inclined to disagree towards the eight defined rationalities of PIM. However, gender and provincial disaggregation of households' responses unfold important differences (Figure 5). Punjab province's gender disaggregated analysis (Figure 5a) reveals that compared to their female counterparts, males tend to agree more with all rationalities. While all female responses are statistically significantly different from male responses in Punjab, the largest gender difference occurs in the responses for governmental rationality followed by financial, organizational, technical, social, economic, political, and environmental rationalities.

On the other hand, Sindh province's gender disaggregated analysis (Figure 5b) is the opposite to the trends observed for Punjab, as females tend to agree more with the rationality statements, with the exception of organizational rationality. Nevertheless, gender mean differences are not as stark as the case of Punjab. While there is no statistically significant difference for economic, financial, and governance rationalities, the largest gender mean difference for Sindh is for political rationality, followed by environmental, economic, technical, social, and financial rationalities (Figure 5b).

**Table 2.** Different rationalities and their questions.

| Rationality | Constructs<br>(Respondents' Agreement/Disagreement under Each Rationality) |
|---|---|
| Technical Rationality | 1. Infrastructure is regularly repaired and well maintained<br>2. Water release/Distribution are scheduled and managed |
| Environmental Rationality | 1. Environmental care and problems are well addressed<br>2. Flooding and flood waters are well controlled |
| Economic Rationality | 1. Adequate infrastructure and marketing/processing arrangements are in place<br>2. Water availability and management lead to good income and profit |
| Social Rationality | 1. Changes bring participation and inclusion of female views<br>2. People/women of all social groups can participate, and hold posts |
| Political Rationality | 1. There is adequate representation of women in leadership roles<br>2. Farmer organizations and Khal Panchayat are able to ensure fairness and justice |
| Organizational Rationality | 1. The water user association (WUA), general bodies and Executive Committees meet regularly<br>2. Leadership/staff is knowledgeable & competent to managing WUA activities |
| Financial Rationality | 1. No mismanagement, Diversion or loss about funds takes place<br>2. The WUA receives sufficient funds and is financially sound |
| Governance Rationality | 1. Government controls and rules are reasonable/good<br>2. Government officials help in planning, mobilizing, organizing, implementation and dispute resolution |

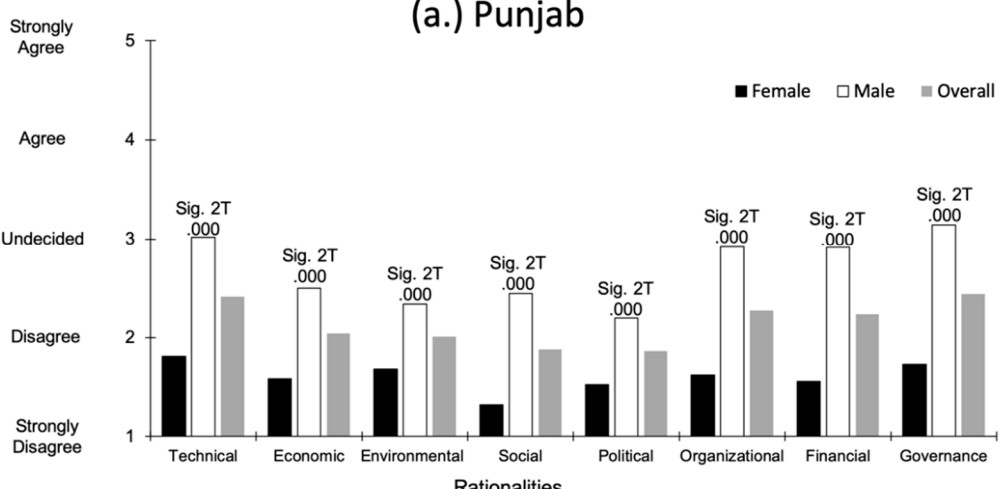

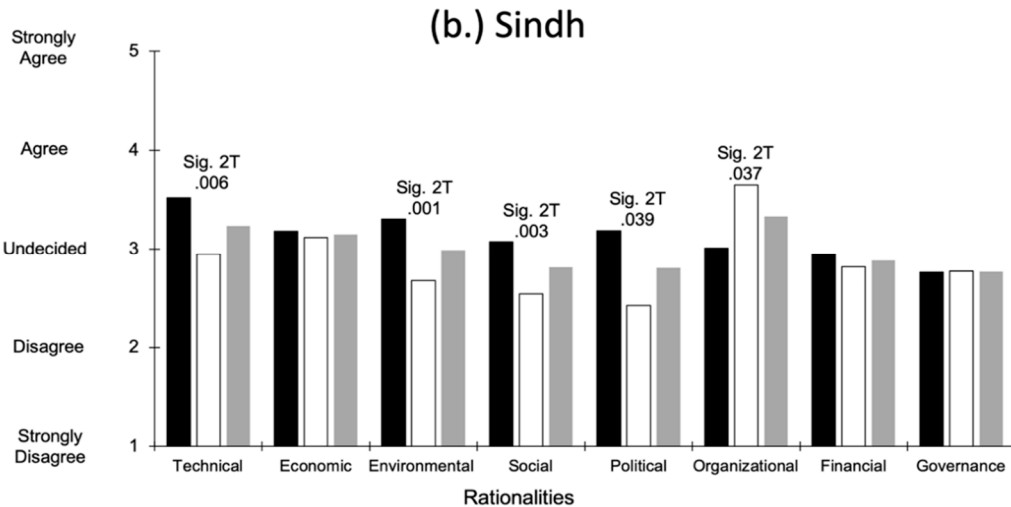

**Figure 5.** Comparative assessment of rationalities for Participatory Irrigation Management.

### 4.3. Comparative Assessment of Institutional Features

Institutions that endure tend to be those whose objectives are clear, opportunities to interact are good, adaptability to changing socio-economic and political conditions are high, scale is appropriate (e.g., neither too small nor too large), and there is a working mechanism to ensure compliance with rules and regulation and to reward and punish decisions [2]. These important markers of institutional assessment were operationalized for PIM institutions in our study areas (see Appendix A, Table A1).

The assessment of households' perceptions of institutional features indicates similar trends as observed from our assessment of rationalities. Females in Punjab are more critical on various institutional features compared not only to their male family members but also to their female counterparts in Sindh (Figure 6a). Although all means of institutional features for females and males in Punjab are statistically different to each other, the highest mean difference appears for adaptiveness followed by clarity of objectives, good interaction, and appropriateness of scale.

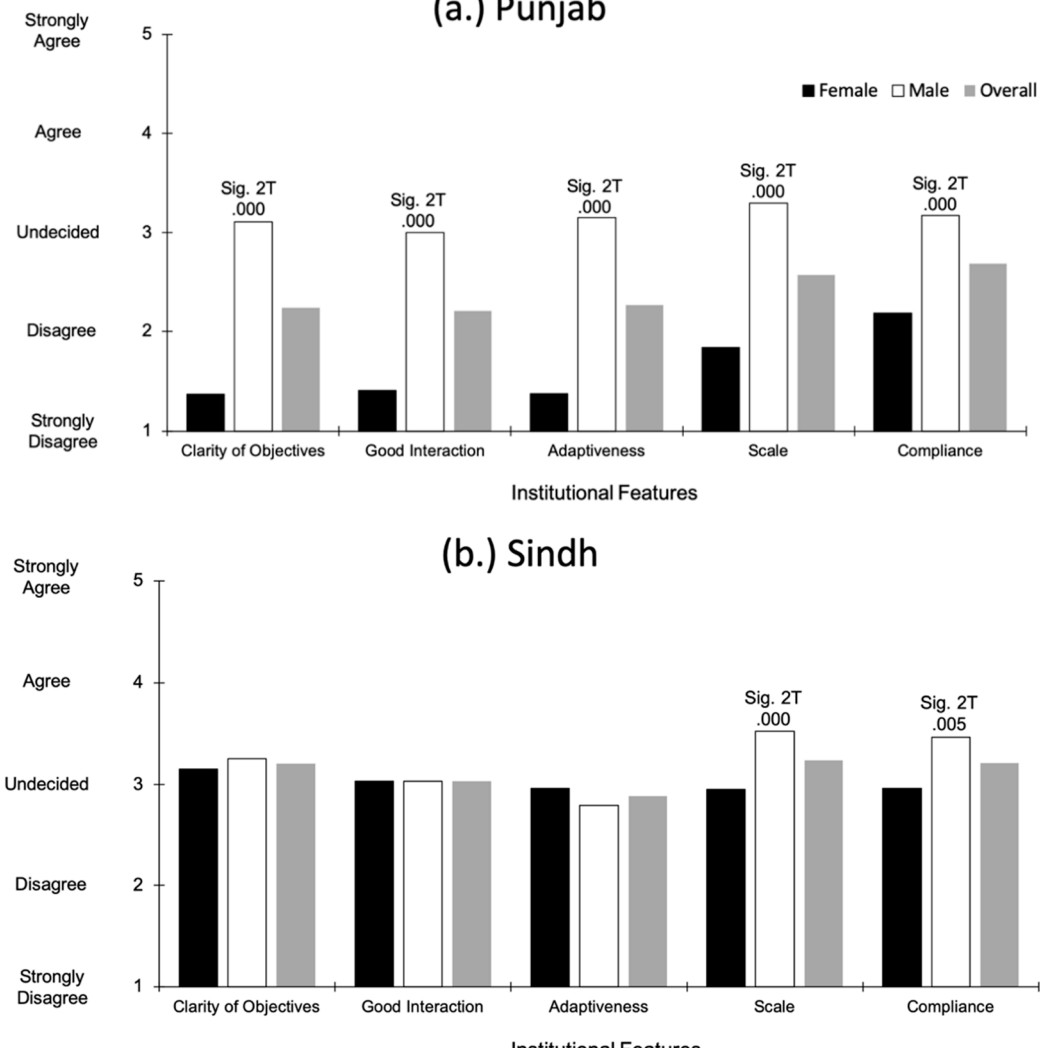

**Figure 6.** Comparative assessment of institutional features of PIM.

Male and female members in Sindh share similar perceptions towards PIM institutional features, except for the questions regarding scale appropriateness and compliance features (where male members are statistically more inclined to agree compared to their female counterparts—Figure 6b).

*4.4. Comparative Assessment of Local Impacts of PIM*

As discussed earlier, PIM institutions aimed to improve water availability, ensure equity in irrigation distribution, resolve key environmental problems such as waterlogging and salinity, improve full-cost recovery, and thereby better financial management—a set of problems that is inherent in the traditional top-down irrigation management system that still prevails in most parts of Pakistan. To assess how PIM may have helped address some of these problems, four major areas and a corresponding set of questions under each were asked of households, as detailed in Appendix A, Table A2.

Our analysis of the local impacts of PIM suggest that inter-provincial and gender-based differences emerge as well, but the respondents are generally undecided (Figure 7). However, female respondents from Punjab are statistically more likely to disagree with any statement about the positive impact of PIM institutions (Figure 7a). It is, however, interesting to note that while there is negligible variance in female's scores assigned to different markers of local impact, men's responses did differ. Men's highest scores were assigned to improved water availability and its impact on production, followed by financial management, environmental management, and equity (Figure 7a).

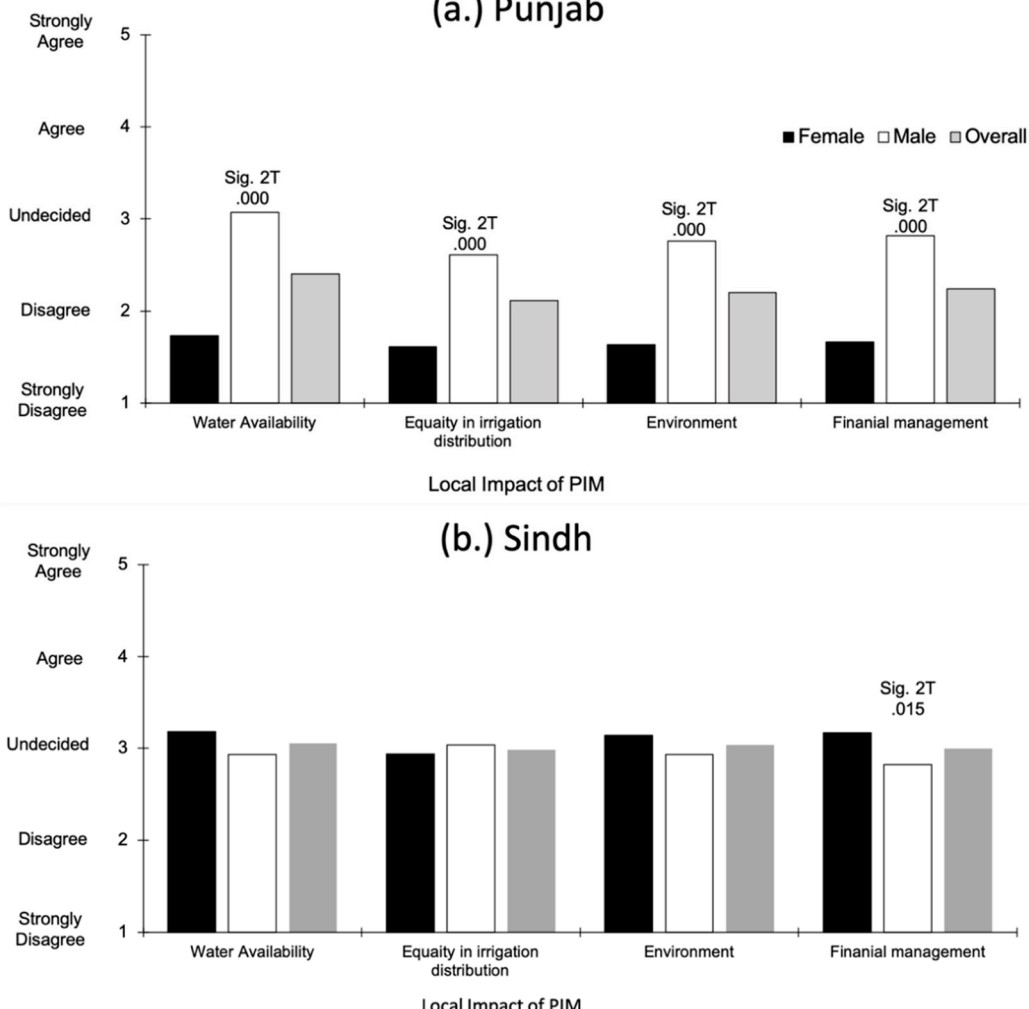

**Figure 7.** Comparative assessment of the specific impacts of PIM.

Female Sindh respondents generally assigned higher scores than their male counterparts, with the exception of the questions on equity impacts (Figure 7b). However, the only statistically significant difference of perception was for financial management.

### 4.5. Comparative Assessment of Wider Impacts of PIM

Society in the rural areas of Pakistan revolves around agriculture and allied activities. Irrigation service and institutions that provide it are one of the fundamentals of all sorts of agricultural activities. Thus, any institutional change in irrigation institutions is likely to have much wider impacts than formally sought through policy interventions like PIM. Men and women were asked their agreement or disagreement with various statements aimed at exploring the wider impacts of PIM (see Appendix A, Table A3).

Vastly different gender results emerge, particularly in the case of Punjab. Men tend to agree more positively than their female counterparts on all wider impact questions (Figure 8). Men, who are generally more active in irrigation affairs and often more visible to irrigation managers and thereby are likely to have better exposure to the way systems work, agree more on questions, such as PIM's impact on the village as a whole, agriculture and allied activities, tail-reach farmers, and youth (age 15–29) years. However, even men strongly disagreed that PIM has had any positive impact on marginalized groups, such as women and lower income/poor groups, in society (Figure 8).

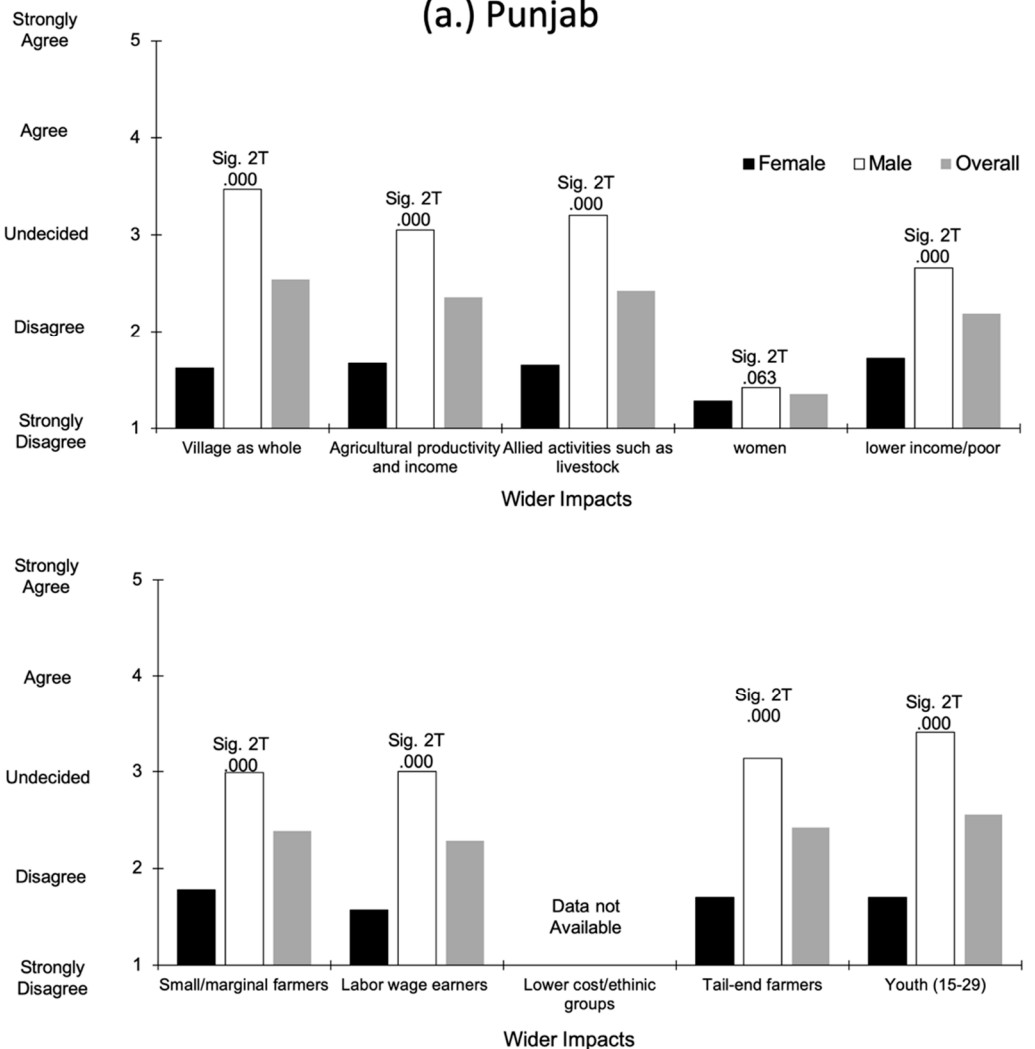

**Figure 8.** Comparative assessment of wider impacts of PIM (a, Punjab).

Other key gender perception differences in Punjab include the impact on the village as a whole, followed by youth, allied agriculture, tail-reach farmers, laborers/wage earners, agricultural production,

small marginal farmers, and low-income poor and women. Both men and women disagree that PIM has had a positive impact on women (Figure 8).

In Sindh, there is a higher likelihood of agreement by both genders for most of the wider impacts of PIM, with women agreeing slightly more on many parameters, though most of these differences were not statistically significant (Figure 9).

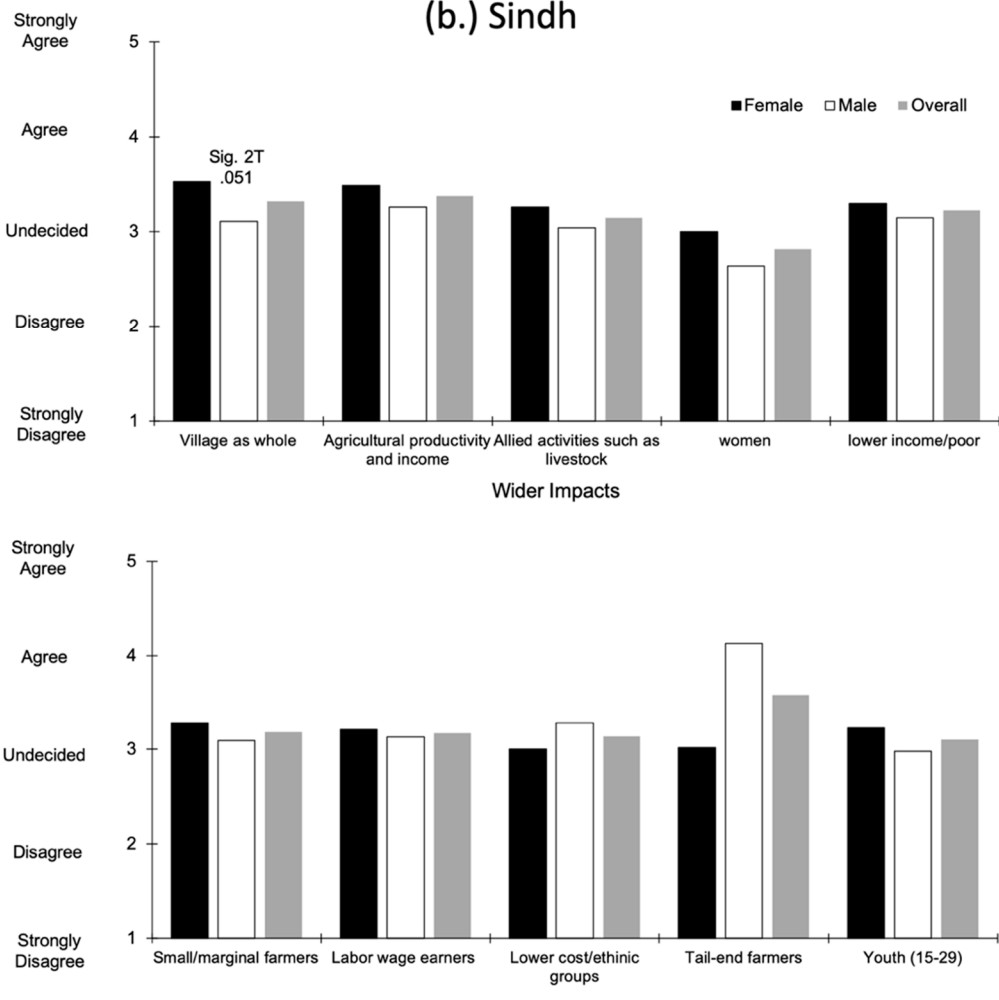

**Figure 9.** Comparative assessment of wider impacts of PIM (b. Sindh).

In Sindh, the Figure 9 results show that there is an overall tendency towards agreement over the entire set of wider impacts of PIM, except in the case of the impact on women (similar to Punjab findings). This suggests that at varying degrees women have been relatively neglected in PIM initiatives in both provinces.

## 5. Discussion

Agriculture in general, and crop-farming in particular, is the dominant occupation in rural areas of Pakistan. Despite their significant role in agricultural activities, women have long remained unseen to many outsiders as rural society was divided along gender lines—undermining females' role and held religious and cultural obligation of purdah responsible for it [17]. This remains the case with irrigation management; because this lies in the public sphere, it is thereby considered a male-only domain in Pakistan. However, researchers [66–70] are increasingly revealing the layers of economic status,

religiosity, culture, and politics of women's work and there are some indications that the situation may be changing slowly [71].

Our study follows on from these earlier studies by highlighting how male and female perceptions on irrigation management vary in Pakistan, using 128 paired matching household face-to-face surveys. Gender perception differences on various parameters were found, such as for a number of rationalities for PIM, institutional features, and the local and wider impacts of overall performance of PIM. Greater gender perceptional differences were particularly found in Punjab; however, female perceptions also differ significantly from their counterparts in Sindh on some PIM aspects. Overall though, in Sindh, men and women share a similar understanding on most aspects of PIM's performance. Although our study is one of the few to highlight gender differences in irrigation in Pakistan, it is important to note that the methodology chosen only provides limited insights into real drivers of male and female preferences across regions. We suggest that future research should concentrate on trying to model these differences in greater detail and understanding the drivers of irrigation management perceptions by gender.

Institutions that last exhibit certain characteristics, and those subject to any institutional experiments must demonstrate various institutional features if long-term sustainability is desired [72]. Previous research on small-scale irrigation and other natural resource systems have revealed institutions endure when those sharing a resource system have clear institutional objectives, strong interaction amongst users, adaptability to changing socio-economic conditions, appropriateness of scale of human organization with reference to physical boundaries of the system, and a mechanism to ensure compliance with management decisions [2,73,74].

Our general results reveal that there is a lack of agreement over different institutional PIM features in Pakistan and respondents either disagree or are undecided with questions about various institutions' features. This may be due to the fact that PIM in Pakistan was a supply-driven policy initiative that was introduced as a result of donor push and there has been limited effort invested in enhancing peoples' understanding of institutional change [9,75].

Stark gender differences in our Punjab surveys imply that more efforts are required to understand the reasons for female perceptions and to address potential inequality issues. Although the interview settings in Sindh were not as confidential as were in the case of Punjab, it is still clear that the province has made some effort in bridging the gender divide in PIM institutions.

This in fact is a reconfirmation of one of the key findings of the World Bank's final report on the "National Drainage Program", where serious effort towards reforming tertiary-level irrigation management, if there was any, was observable only in the Sindh province [75]. Besides these recent efforts in women's social mobilization on account of PIM implementation in the province, there is ample historical evidence that rural women in Sindh are politically and socially more aware of the activities taking place in the surrounding 'male domains' than their counterparts in Punjab. Their experiences with socio-political mobilization struggles under the banners of movements, such as Sindhiani Tehrik, may have equipped females in Sindh more so than in Punjab to critically evaluate the transformative potential of PIM.

Even in the case of the Sindh region, where significant investment has been made in implementing PIM, there are still many practical problems hindering gender equality. Initial efforts were focused on forming separate women's groups but without any formal utility in water management decisions. This caused these groups to gradually become redundant. One of the important causes of the failure of such a group was inadequate social mobilization. For example, the Sindh Irrigation and Drainage Authority always used to hire female social mobilizers to work with women and mainstream them in the reform program, but in reality, they were never able to adequately mobilize such staff in the field due to the unavailability of vehicles and other logistic support. Nowhere in the minutes of the meetings of any farmer organization can women's meaningful and agenda-driven representation be found. However, even with such limited social mobilization, some females were reported being

the office bearers of farmer organizations, but this has never occurred to such an extent where their presence can be considered as representative [9].

At the same time, major institutional and infrastructure development initiatives like the Sindh Water Sector Improvement Program or the Sindh Barrages Improvement Programs started working on women's differential water needs and constructed structures, such as washing ghats, near villages to help them perform their traditional gender roles. Although we think this is not necessarily the most ideal way to include gender representation, these efforts may have increased female views towards participatory water management programs.

Putting these findings in the rich historical context developed earlier in this paper explains why PIM's acceptance, particularly among females, has been relatively higher in Sindh compared to Punjab. Better political activism of men and women in Sindh and the people-centric connotation of the reform may have created a general goodwill and compelled the government institutions (such as SIDA) to improve PIM legislation and implementation; whereas the lack thereof has caused a roll back of an already limited rise of these participatory institutions in Punjab.

## 6. Conclusions

This study sought to provide insight into the effectiveness of inclusion in the context of irrigation management in two areas in rural Pakistan by providing empirical comparisons between the perceptions of women and men farmers towards PIM. We found that there was often a significant difference in perceptions across gender and also across the two study jurisdictions. Notably, women generally perceived the performance and impact of farmer organizations to be significantly less effective than men do. Accordingly, the preferences of the included and excluded should not be treated as homogenous. Moreover, both men and women do not support the view that PIM has had a positive impact on women. This highlights the challenge of achieving inclusiveness in general. Importantly, historical choices about inclusive and exclusive roles play a significant role and the findings must be considered against this path dependency.

A historical account of women's struggle entails that achieving their effective involvement in the current context is not without significant challenges and complexities. A path forward would need to move beyond the latitudes of legislation that create women's groups that do not take into account actual female involvement. Notwithstanding the tokenistic efforts to incorporate women into water management and the interest from women themselves to be involved, there needs to be more consideration given to the type of real involvement that would lead to improved livelihoods and improved decision-making for all. Essentially, the results of this study suggest that effective involvement requires a complete understanding of the perspectives of women in their historical context rather than assumptions about their preferences for the type and extent of their involvement.

**Author Contributions:** Conceptualization, J.A.M., B.C. and S.W.; methodology, J.A.M., B.C. and S.W.; formal analysis, J.A.M.; investigation, J.A.M.; writing—original draft preparation, J.A.M. and B.C.; writing—review and editing, J.A.M., B.C. and S.W.; supervision and final editing, J.A.M., B.C. and S.W. All authors have agreed to the final version.

**Funding:** Australian Centre For International Agricultural Research (ACIAR), funded this study through a larger research project titled Efficient Participatory Irrigation Institutions to Support Productive and Sustainable Agriculture in South Asia [ADP/2014/045].

**Acknowledgments:** Field data collection support from Muhammad Ashfaq and his team at University of Agriculture Faisalabad, Bakhshal Khan Lashari and his team at U.S.-Pakistan Center for Advanced Studies in Water (USPCAS-W) Jamshoro and overall supervision of Bashir Ahmad of Pakistan Agriculture Research Council, Islamabad were much appreciated. We also acknowledge the insightful feedback and encouragement of three anonymous reviewers who helped us substantially improve this paper.

**Conflicts of Interest:** The authors declare no conflict of interest.

## Appendix A

**Table A1.** Different institutional features and their constructs.

| Institutional Features | Constructs (Respondents' Agreement/Disagreement under Each Institutional Feature) |
|---|---|
| Clear Objectives | 1. The WUA objectives/purpose/roles are known and clear to everyone<br>2. The WUA regularly makes plans and actions towards achievement of the objectives |
| Good interaction | 1. There is good leadership to help and guide the interactions<br>2. Regular & frequent meetings with participation of all social and farmer groups |
| Adaptiveness | 1. Rules, plans and procedures are sensitive to member needs and local conditions<br>2. There are instances when the rules were changed to meet local conditions |
| Scale/Size | 1. The scale/size/scope of operation of the WUA is appropriate<br>2. The current distribution of powers, resources & responsibility between government, WUA and farmers is appropriate |
| Compliance | 1. The WUA's rules and schedules for water management frequently complied by members/villagers<br>2. The WUA uses its powers to bring compliance to the rules |

**Table A2.** Impact of participatory irrigation management institutions.

| PIM Benefits | Constructs (Respondents' Agreement/Disagreement under Each Institutional Feature) |
|---|---|
| Water management Benefits | 1. Greater water availability<br>2. Timely water availability<br>3. Better and timely maintenance and repairs<br>4. Higher Incomes |
| Equity Benefits | 1. Wider membership and greater involvement<br>2. Greater sense of ownership and empowerment of farmers<br>3. Greater involvement and empowerment of women<br>4. Greater fairness and justice |
| Environmental Benefits | 1. Better care of the environment and biodiversity<br>2. Better conservation of water<br>3. Better conservation of soils, reduction in soil erosion<br>4. Reduction in flood damage |
| Financial Management Benefits | 1. Better collection of fees and charges<br>2. Better availability of funds and support from the government<br>3. Better financial discipline and avoiding misuse of funds<br>4. Greater financial strength |

**Table A3.** Wider impact of participatory irrigation management institutions.

| Constructs (Respondents' Agreement/Disagreement with Wider Impacts of PIM) |
|---|
| 1. Wider impact on village as a whole |
| 2. Wider impact on agriculture productivity and incomes |
| 3. Wider impact on allied activities (e.g., livestock, fisheries) |
| 4. Wider Impacts on women |
| 5. Wider Impacts on labor/wage earners |
| 6. Wider Impacts on lower income/poor |
| 7. Wider Impacts on small/marginal farmers |
| 8. Wider Impacts on lower caste/ ethnic groups |
| 9. Wider Impacts on tail-reach farmers |
| 10. Wider Impacts on youth (15–29 Years) |

**Table A4.** Overall assessment of PIM institution's performance.

| Constructs<br>(Respondents' Agreement/Disagreement on Overall Assessment) |
| --- |
| 1. WUA's overall performance<br>2. Water availability and related economic/income benefits<br>3. Equity in water distribution, and its benefits<br>4. Environmental impact and outcomes<br>5. Financial management and control |

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
