# Peer review of "Mainstreaming Gender into Irrigation: Experiences from Pakistan"

_water, doi:10.3390/w11112408_

Round 1

Reviewer 1 Report

The manuscript can be published in the present form. 

Author Response

Comment:

The manuscript can be published in the present form. 

Response:

Many thanks for your review.

Reviewer 2 Report

This paper addresses gender issues in participatory irrigation management in Pakistan. The topic is relevant and deserves attention. However, in my opinion, a deeper and better structured analysis would be needed to put in value the strengths of the study.

For instance, the topic is presented in the summary. However, the summary should also point to the main conclusions of the study. Also, while the focus is on gender issues, no keyword is provided on that topic. Therefore, the list of keywords should be reviewed.

On the background, I miss a description of the study area. The focus on the background is on gender issues and not on irrigation, which should also be covered. To understand the implications of this study, we need to know what the irrigation reality is in the Sindh and Punjab regions. How irrigation has evolved through time? What participatory irrigation initiatives have been implemented? What other relevant studies on participatory irrigation exist?

I find it highly relevant the idea of analysing male and female perceptions. This is an interesting exercise, but the authors should draw more insightful conclusions from their analysis. For instance, local impacts of PIM do not seem to be high. What are the main reasons and how this fact influences the gap between male and female perceptions? Or, for instance, how perceptions compare to some objective performance indicators?

On another level, I find many grammatical errors throughout the text. A professional proofreading is needed to ease readiness. Furthermore, symbols used should be homogenised throughout the text. For example, in Figure 3, male answers are labelled with a black square and, in Figure 4 the black square corresponds to female answers. In addition, labels in Figures have to be checked (i.e., there are several spelling errors in Figure 6).

Author Response

Comment 1:

This paper addresses gender issues in participatory irrigation management in Pakistan. The topic is relevant and deserves attention. However, in my opinion, a deeper and better structured analysis would be needed to put in value the strengths of the study.

Response:

We agree there are several possible ways to analyze this data. However, the method was specifically chosen to provide insight into our key research questions. This is supported by the comments from Reviewer 1 & 3 who recommended publication of this paper in its present form.

We value this feedback and think it is a useful consideration for further research, however we believe it is beyond the scope for this particular paper.

However, as for the more specific comments of Reviewer 2 are concerned, we have incorporated these to the best way we could.

Comment 2:

For instance, the topic is presented in the summary. However, the summary should also point to the main conclusions of the study. Also, while the focus is on gender issues, no keyword is provided on that topic. Therefore, the list of keywords should be reviewed.

Response:

Agreed, we have revised our abstract: Introductory lines in the abstract are shortened and key conclusion of the study are now included. See lines 21-26.

We have reviewed the keyword list following this useful suggestion and have added two new keywords: Gender analysis and Inclusion.

Comment 3:

On the background, I miss a description of the study area. The focus on the background is on gender issues and not on irrigation, which should also be covered. To understand the implications of this study, we need to know what the irrigation reality is in the Sindh and Punjab regions. How irrigation has evolved through time? What participatory irrigation initiatives have bee implemented? What other relevant studies on participatory irrigation exist?

Response:

Agreed, we have added a comprehensive description of the study area, in both the background and a few details in the method section as well. For example:

Following two sections have been added besides a new figure (Fig 1) has been included which shows considerable detail of the irrigation system and study area. 

2.3. Evolution of the Indus River’s Irrigation Network

2.4. Water resources management in the Indus River Basin

In these two sections, we have provided an outline of the irrigation in Sindh and Punjab regions to address the reviewer’s questions  such as how How irrigation has evolved through time? What participatory irrigation initiatives have been implemented? What other relevant studies on participatory irrigation exist?). Regarding the latter, you would notice that reference list has grown by at least 13 new references. We have also removed to 2 duplicate references from the list.

In doing so, we have also revised Fig 2 to accommodate a very recent legislative development on the subject.

Comment 4:

I find it highly relevant the idea of analysing male and female perceptions. This is an interesting exercise, but the authors should draw more insightful conclusions from their analysis. For instance, local impacts of PIM do not seem to be high. What are the main reasons and how this fact influences the gap between male and female perceptions? Or, for instance, how perceptions compare to some objective performance indicators?

Response:

In addition to our response to comment 1, we again agree that much more can be done with the data, but, we believe that to do so would require a different set of analysis (something like structural equation modelling for example). Given this would require basically a new paper, with a different methodological orientation, it was not possible for us to tackle it in this paper.

However, we have noted both a) the limitations of our study and b) the need for further research in our paper. See Lines 508-513. 

Reviewer 3 Report

This looks like a nicely written, structured, and though-through manuscript. It provides original empirical evidence to substantiate the argument, and therefore I would recommend it for publication. I particularly liked its clarity, and the presentation of the data, as well as the justification and richness of the data used.

Author Response

Comment:

This looks like a nicely written, structured, and though-through manuscript. It provides original empirical evidence to substantiate the argument, and therefore I would recommend it for publication. I particularly liked its clarity, and the presentation of the data, as well as the justification and richness of the data used.

Response:

Many thanks for your review.

In addition, we have done following to further improve the presentation of the paper:

Figure 3 and Figure 9 have been replaced to make their formatting consistent with rest of the figures.

We have proof-read and edited the paper again to improve its clarity and presentation.